# Staple Food Item Availability among Small Retailers in Providence, RI

**DOI:** 10.3390/ijerph16061052

**Published:** 2019-03-23

**Authors:** Yuyao Huang, Alison Tovar, John Taylor, Maya Vadiveloo

**Affiliations:** 1Department of Nutrition and Food Sciences, University of Rhode Island, Kingston, RI 02881, USA; yuyao_huang@my.uri.edu (Y.H.); alison_tovar@uri.edu (A.T.); 2Department of Plant Sciences and Entomology, University of Rhode Island, Kingston, RI 02881, USA; jr_taylor@uri.edu

**Keywords:** food environment, food desert, SNAP, food assistance, community, local food culture, nutrition, small food stores, food insecurity

## Abstract

Inventory requirements for authorized Supplemental Nutrition Assistance Program (SNAP) retailers have undergone several revisions to increase the availability of healthful foods. A proposed rule of 84 staple food items was not implemented due to concerns that stores would not withstand this expansion, resulting in a final rule requiring 36 items. This study used the Food Access Research Atlas data to characterize food provisions in 30 small retailers in areas with high and low proportions of SNAP and racial minority residents in Providence, Rhode Island (RI). Stores were assessed with an audit instrument to tally variety, perishability, and depth of stock of four staple food categories. Descriptive, analysis of variance, and chi-square analyses were performed. Across stores, 80% were compliant with the final rule, but 66.7% would need to expand their offerings to meet the proposed rule. Mean dairy variety was lowest among all categories (*p* < 0.05). Most stores met the perishability (92.3%) and depth-of-stock requirements (96.1%) under both rules. No difference was detected between areas with high and low proportions of SNAP and racial minority residents. Future expansion of requirements may increase healthful food availability without imposing undue burdens on retailers in Providence, RI, excluding increased requirements for dairy variety.

## 1. Introduction

In the United States, the prevalence of obesity among adults increased linearly in the periods of 1999–2000 and 2015–2016 (30.5%–39.8%) [1]. In particular, obesity rates are disproportionally higher among Hispanic (47.0%) and non-Hispanic black (46.8%) adults, compared to other racial/ethnic groups [1]. Additionally, 21.2% of non-Hispanic black and 18.3% of Hispanic households live under the federal poverty level nationwide [2], which could exacerbate obesity rates, given the proportional relationship between obesity and poverty rate [3]. Low socio-economic (SES) populations (i.e., populations with lower income and lower education level) also have lower diet quality than high-SES populations, and the difference in the Alternate Healthy Eating Index 2010 score between high-SES and low-SES groups increased substantially 3.9–7.8 from 1999–2000 to 2009–2010, with non-Hispanic black adults having the lowest diet quality among all ethnicities [4]. One contributor to their lower diet quality and subsequently higher obesity prevalence may be limited access to healthful foods [5,6,7,8]. According to a systematic review in 2010 [9], lower-income neighborhoods have fewer food outlets and are less likely to provide healthful food options to residents. While low-income households with automobiles often travel beyond the nearest stores to shop for food [10], 2.3 million households in the U.S. do not have access to a vehicle or public transit, and are estimated to live more than a mile from a supermarket [11]. Low-income and predominantly black and Hispanic neighborhoods have fewer supermarkets, and blacks, on average, travel a greater distance to reach their nearest supermarket [9]. Areas with limited food access, often termed “food deserts”, commonly exist in low-income neighborhoods with predominantly black and Hispanic populations [9,12,13,14]. Residents in these underserved neighborhoods may benefit from increased selections of foods in the limited number of small retailers readily accessible to them.

One program intended to improve diet quality among those experiencing food insecurity is the United States Department of Agriculture (USDA) Supplemental Nutrition Assistance Program (SNAP), which subsidizes foods consumed at home, excluding alcohol, supplements and medicines, and hot foods [15]. Recently, SNAP proposed expanding stores’ inventory requirements in an effort to improve healthful food access for low-income populations, among whom 42 million participate in SNAP [16]. While SNAP receipt has reduced food insecurity [17,18], participation is associated with a higher Body Mass Index (BMI), waist circumference, and metabolic risk factors [19], and SNAP participants have higher obesity rates than income-eligible and higher-income nonparticipants (40% versus 32% and 30%, respectively) [20]. Additionally, while the evidence is not conclusive, the general consensus is that SNAP recipients have lower diet quality compared to higher-income non-participants, and that most low-income populations, regardless of SNAP participation, fail to meet the Dietary Guidelines for Americans [21,22].

To encourage consumption of healthful foods among SNAP participants, public health efforts have promoted multiple interventions that aim at improving food purchasing behavior. One successful effort which was instrumental in informing the development of SNAP inventory requirements was expanding the provision of healthful foods in small food retailers [23,24,25,26,27,28]. Several attempts have been made at increasing the variety and quantity of staple foods that SNAP-authorized stores are required to carry. According to the USDA Food and Nutrition Service (FNS), staple foods are defined as basic food items that make up a significant portion of an individual’s diet and are usually prepared at home and consumed as a major component of a meal [29]. This broad definition allowed for certain food items that could be regarded as accessory foods to be counted toward staple foods, such as cream cheese, cream, and butter. The original rule from the Food Stamp Act of 1977 [30] required stores to carry a minimum of three varieties under each of the four staple food categories: fruits and vegetables; meat, poultry, and fish; breads and cereals; and dairy, including at least two categories with perishable items (with a minimal stocking of 12 items). In February 2016, a new rule was proposed to increase the number of required staple items to 168 (a minimum of seven varieties in four categories, plus six depth-of-stock, i.e., the minimum number of units of every variety). However, depth-of-stock was reduced to three (84 items in total, i.e., 4 categories × 7 varieties × 3 depth of stock) in December 2016 (hereafter referred to as the proposed rule), due to concerns that the expansion would impose a substantial burden on small retailers [31]. One common concern was that the increase in the number of perishable items would result in spoilage and waste, the cost of which is non-refundable by manufacturers. Another concern was stocking logistics, such as a larger quantity of food items exceeding the available shelf space [31]. Others expected that it would be difficult for retailers to carry the proposed seven varieties in the meat, poultry, and fish category and the dairy category, and in response to this particular concern, the proposed rule incorporated plant-based products as eligible varieties in these two categories [31]. However, the proposed rule was not implemented, due to the Consolidated Appropriations Act in 2017 that required revision of the definition of “variety” [32]. The final rule, which became effective in January 2018, reinstated the requirements of the original rule, plus a minimum depth of stock of three (36 items in total). (See Appendix A and footnotes of Table 3 for details.)

The idea that the proposed expansion of the rule is unacceptable stems from the concerns of an increased administrative burden and profit loss for smaller retailers [33,34,35,36]; however, current knowledge about whether smaller retailers would adapt to the more stringent inventory requirements is limited. Therefore, it is important to evaluate how well they can meet the different requirements to help determine whether the concerns related to cost and burden are well-founded. In addition, the perishability of foods within each category was also explored (i.e., whether a store carries a variety of foods by having predominantly perishable items or shelf-stable ones), as it has been reported that SNAP participants with limited access to perishable foods, particularly fruits and vegetables, have lower expenditure on those foods, which are part of a healthful dietary pattern [21,37].

Therefore, this study set out to characterize the food availability of 30 small retailers (approximately 73% of total small retailers) in five census tracts identified as food deserts in Providence, a city in the state of Rhode Island (RI), aiming to help inform future inventory rules. This study stemmed from a multi-site project conducted by the Illinois Prevention Research Centers (PRC) Nutrition and Obesity Policy Research and Evaluation Network (NOPREN) Collaborating Center [38]. Specifically, the following questions were examined: (1) What proportion of the retailers met the different requirements, including variety, perishable categories (hereafter referred to as perishability), depth-of-stock, and total stocking in the original rule, final rule, and proposed rule, respectively; (2) On average, how many varieties, varieties of perishable items, depth-of-stock, and total stocking did stores carry in each of the four staple food categories; (3) What proportion of the retailers carried perishable items within each category (though this was not a requirement in any of the rules); and for all three questions, we compared whether there was a difference between areas with high and low proportions of SNAP and racial minority populations.

## 2. Materials and Methods

### 2.1. Overview

A total of 30 stores were sampled from selected food deserts in Providence, RI. A five-page audit instrument was used to collect information at each store. To assess the store inventory status under the original rule prior to the implementation of the current rule in January 2018, data collection took place in November and December 2017, skipping the week before Thanksgiving and the week of Thanksgiving, due to a potential inventory change for the holiday.

### 2.2. Sampling

Because of the important role small retailers play in healthful food purchases in low-income neighborhoods, this study focused on assessing small food stores in selected food deserts in Providence, where the population is 34.3% non-Hispanic white, 42% Hispanic, and 15.6% non-Hispanic black [39], with 26.9% of residents living under the poverty level, higher than the national average of 12.3% [2,39]. A food desert, as defined by the USDA Economic Research Service (ERS) [40] is a census tract that is low-income (LI) and low-access (LA), with low-income residents and few large food stores. (See definitions of census tracts, LI, and LA in Appendix B [40,41].) In order to locate food deserts, ERS developed the Food Access Research Atlas [42], an online database that assembles indicators of the food environment, such as access and proximity to food stores at the census tract level. This tool has been employed in many previous studies investigating food retail access [43,44,45]. Tracts were selected by layering the Food Access Research Atlas LILA tracts on Google Maps and 2010 Census Tracts Shapefiles [46] and merging them with Food Access Research Atlas 2015 data [47] in the Quantum Geographic Information System (QGIS) [48], an open source mapping tool. Tracts in Providence where at least 500 people or at least 33 percent of the population is greater than ½ mile from the nearest supermarket, supercenter, or large grocery store (hereafter referred to as LILA areas) were identified as food deserts for this study. Seventeen out of 39 census tracts in Providence, RI were identified as LILA areas, among which five tracts were further selected based on the characteristics of interest shown in Table 1. Three tracts with a low proportion of non-Hispanic black and Hispanic, and a low proportion of housing units receiving SNAP benefits were selected (hereafter referred to as low SNAP and low racial minority tracts, mean = 31.5% and 37.6%, respectively). Two tracts with a high proportion of non-Hispanic black and Hispanic, and a high proportion of housing units receiving SNAP benefits were selected to serve as a comparison (hereafter referred to as high SNAP and high racial minority tracts, mean = 2.8% and 3.1%, respectively).

Small non-chain grocery stores, convenience stores, dollar/discount stores, pharmacies/drug stores, and liquor stores, irrespective of SNAP authorization status, were included to obtain a comprehensive assessment (inclusion criteria). Given the focus of this study on small retailers, supermarkets and chain grocery stores were not assessed. Farmer’s markets, butcher shops, and bakeries were also not assessed as they are not required to meet the inventory requirements to be SNAP-authorized [29] (exclusion criteria). (Definitions of each type of store can also be found in the Congressional Research Service (CRS) Report R44650 [49].) Stores that met inclusion criteria were identified from each tract using Google Maps and the USDA SNAP retailer locator [50]. A total of approximately 41 small retailers were identified among the five tracts. When a store was unavailable for auditing (i.e., did not exist or had permanently closed) or when store owners raised objections, a new store was added to the list of stores in order to obtain the desired sample size of 30 stores, with 15 stores in high SNAP and high racial minority tracts and 15 stores in low SNAP and low racial minority tracts (Table 2, Figure 1).

### 2.3. Retail Audit

An Institutional Review Board exemption was obtained prior to store auditing. (ID: 1139592-1) All stores were accessed using an audit instrument developed by the Illinois Prevention Research Centers (PRC) and Nutrition and Obesity Policy Research and Evaluation Network (NOPREN) Collaborating Center [38], which provides measures for store characteristics (e.g., interior and exterior features) and the availability of food items in the store (Appendix C). The instrument has been tested for reliability in Illinois PRC research [38]. Auditors recorded information on up to 10 foods and/or beverage varieties under each of the four staple food categories, in the order of the fruits and vegetables category, to the meat, poultry, fish category, to the breads and cereals category, and lastly to the dairy category. Under each category, food items were searched for from left to right in the sequence shown on the instrument page, giving priority to perishable items. Once 10 varieties were identified for a given category and at least one perishable item had been identified, that category was considered complete, and no additional data were collected. Accessory foods that are generally considered drinks, condiments, snacks, or desserts were not counted as per the instructions. Food items that were in storage and not available on the shelves at the time of auditing were also not counted. Mixed dishes or multiple-ingredient foods and beverages where the first ingredient was a staple food item were counted toward that accordant category, unless the first ingredient was water, in which case they were classified by their second ingredient. For example, a frozen pizza would count toward the breads and cereals category if flour was the first ingredient, but would count toward the dairy category if the first ingredient was cheese [51]. (Refer to CRS Report R44650 for detailed definitions of accessory foods and mixed dishes [49].) The depth-of-stock requirement under the first proposed rule of 6 was not evaluated as the number of items that could be recorded with the audit tool was limited to 1, 2, or 3, or more. Auditors completed a 2-hour live webinar provided by the Illinois PRC NOPREN Collaborating Center that instructed in detail the use of the audit tool. The webinar was taped for reference, and any questions were answered either during the webinar or afterwards via email communication. A training manual was also provided for reference, with detailed definitions of store type, category, variety, perishability, depth of stock, accessory foods, mixed dishes, and instructions on what should and should not be counted. Two trained auditors assessed the stores together to enhance data quality. Data collection was unannounced, but an explanatory letter written in English and Spanish was provided to store owners and staff when concerns were raised.

### 2.4. Analysis

Based on the multi-site project proposed by the Illinois PRC NOPREN Collaborating Center, a sample size of 30 stores was selected. Because this study was primarily descriptive, a priori power was not calculated. Post hoc power was computed to evaluate the difference between high and low SNAP/high and low racial minority tracts, with a large effect size of 0.5, α of 0.05, and degree of freedom of 1, resulting in a power of 0.8. The exposure variable for this analysis was stores located in census tracts with a high (versus low) proportion of SNAP and racial minority households. Outcome variables included variety, perishability, depth of stock, and total stocking (see definitions of each in footnotes of Table 3).

### 2.5. Variety

To determine the variety of foods that a store carried, the number of unique items in each of the four staple categories (i.e., fruits and vegetables; meat, poultry, and fish; breads and cereals; and dairy) were assessed, and binary indicators were created to show whether a store met the variety requirement of at least 3 or 7 varieties.

### 2.6. Perishability

To determine the variety of perishable items under each category, foods that are considered perishable (i.e., fresh/frozen/dips/mixed dishes in the fruits and vegetables category; perishable 100% fruit juice or vegetable juice; fresh/frozen/mixed dishes in the meat, poultry and fish category; fresh/ frozen/mixed dishes in the breads and cereals category; and fresh/frozen dairy) were counted. Binary indicators were created to indicate whether a store met the perishability requirement of having a perishable item in at least 2 or 3 categories, if a store carried at least one perishable item under each category, and if a store carried at least 50% perishable items.

### 2.7. Depth of Stock

Binary indicators were created to indicate whether a store met the depth-of-stock requirement of at least 3 units of items in all categories in the proposed rule and final rule.

### 2.8. Total Stocking

Binary indicators were created to indicate if a store met the total stocking requirement of the different rules (*n* = 12, 36, and 84 total items under the original, final, and proposed rules, respectively).

All analyses were conducted using Statistical Package for Social Sciences (SPSS) version 25, developed by International Business Machines Corporation, registered in many jurisdictions worldwide [52]. Descriptive statistics were used to determine the mean and/or proportion of retailers that met the different versions of requirements (i.e., variety, perishability, depth of stock) and the perishable items within each category. One-way analysis of variance (ANOVA) was performed to determine if mean variety differed across categories. Chi-square tests were used to determine whether differences in the means and/or proportions in variety, perishability, and total stocking were significant between high and low SNAP/high and low racial minority tracts.

## 3. Results

Among the 30 small retailers sampled, 80% were non-chain grocery stores and convenience stores; the remaining stores included 3.3% small discount stores, 13.3% pharmacies, and 3.3% liquor stores. (Table 2) Of all stores, 86.7% were SNAP authorized; 6.7% of stores in high SNAP and high racial minority tracts and 6.7% of stores in low SNAP and low racial minority tracts were not SNAP authorized. (Table 2)

### 3.1. Variety

Table 3 shows the proportion of stores that were able to meet the requirements in the original, proposed, and final rule, and also compares whether the store inventory in high and low SNAP/high and low racial minority tracts differed with respect to meeting these requirements. Of all stores, 83.3% met the variety requirement in the original and final rule; that is, three varieties per category, but only 33.3% met the proposed rule, or seven varieties per category. Among SNAP-authorized stores (*n* = 26), 92.3% met the original and final rule, and 38.5% met the proposed rule. No significant difference in variety was detected between stores in high and low SNAP /high and low racial minority tracts; most stores in both high and low SNAP /high and low racial minority tracts met the original and final rules (86.7% and 80%, respectively). Among SNAP-authorized stores, 93.3% in both high and low SNAP /high and low racial minority tracts met the original and final rules. However, fewer met the variety requirements in the proposed rule (i.e., of all stores, 26.7% were in high SNAP and high racial minority tracts and 40% in low SNAP and low racial minority tracts; and among SNAP-authorized stores, 30.8% were in high SNAP and high racial minority tracts and 46.2% in low SNAP and low racial minority tracts).

Table 4 shows the mean of the variety, variety of perishable items, depth of stock, and total stocking in each category. The variety requirement was met the least often in the dairy category (83.3% and 40% that carried 3 and 7 varieties, respectively) compared to the other three categories (90% and 73.3–86.7 that carried 3 and 7 varieties, respectively). On average, stores carried a lower variety of dairy foods (mean = 5.40 ± 3.04) compared to the other three categories (mean = 7.80 ± 3.31 - 8.53 ± 3.16, *p* < 0.05) (Table 4). Among SNAP-authorized stores, 100%, 96.2%, and 80.8% reached or exceeded 3, 7, and 10 varieties in the fruits and vegetables category; 100%, 84.6%, and 50% in the meat, poultry, and fish category; 100%, 84.6%, and 61.5% in the breads and cereals category; and 92.3%, 46.2%, and 3.8% in the dairy category, respectively. SNAP-authorized stores carried more varieties of staple foods (mean = 33.15 ± 6.21) than stores that were not SNAP-authorized (mean = 6.25 ± 9.84, *p* = 0.01).

### 3.2. Perishability

Perishability requirement: Of all stores, 83.3% met the perishability requirement (92.3% among SNAP-authorized stores) in both the original and final rule—that is, at least two categories with perishable items—and the proposed rule—that is, at least three categories with perishable items (Table 3). No significant difference in meeting the perishability requirement was detected between stores in high and low SNAP /high and low racial minority tracts (i.e., of all stores, 80% in high SNAP and high racial minority tracts and 86.7% in low SNAP and low racial minority tracts; and among SNAP-authorized stores, 73.3% in high SNAP and high racial minority tracts and 86.7% in low SNAP and low racial minority tracts) (Table 3).

Perishable items within each category: There was a lower variety of perishable items in the breads and cereals category (mean = 2.07 ± 1.70) compared to other categories (mean = 4.67 ± 2.58 ~ 6.20 ± 4.09, *p* < 0.01) (Table 4). Table 5 shows perishable items within each category. Of all stores, 76.7% had at least one variety of perishable item under all categories, but few carried more than half of perishable items.

### 3.3. Depth of Stock

Depth of stock was not required in the original rule. Only one store did not meet the depth-of-stock requirement in the proposed and final rule (i.e., three or more items per variety) under the fruits and vegetables category and breads and cereals category (Table 3). No significant difference in depth of stock was detected between stores in high and low SNAP/high and low racial minority tracts (Table 3).

### 3.4. Total Stocking

Overall, 80% of all stores were compliant with the total stocking requirement (including categories, varieties, and depth of stock) in the original and final rule, but 66.7% would need to expand their offerings to meet the total stocking requirement in the proposed rule (Table 3). Among SNAP-authorized stores, 88.5% were compliant with the total stocking requirement in the original and final rule, but 61.5% would need to expand their offerings to meet the total stocking requirement in the proposed rule (Table 3). No difference was detected between stores in high and low SNAP/high and low racial minority tracts in meeting the total stocking requirement of the original or final rule (80% in both tracts), as well as that of the proposed rule (53.3% in high SNAP and high racial minority tracts and 80% in low SNAP and low racial minority tracts) (Table 3).

## 4. Discussion

This study set out to measure food availability among small retailers and whether disparities exist for minority populations in census tracts classified as food deserts in Providence, RI, with the aim of helping to inform future stocking rules for SNAP-authorized stores in low-income low-access (LILA) areas as defined in Appendix B, with ½ mile used as the indicator of low access. Overall, most stores met all requirements—including variety, perishability, and depth of stock—in the original and final rules, and a higher number of stores would have also met all requirements in the proposed rule if the dairy variety requirement had been excluded. No disparity was observed between high and low SNAP/high and low racial minority tracts in meeting the different requirements in all categories in the original, final, and proposed rule, or in perishability within categories. Detailed findings are discussed below.

### 4.1. Variety

Among SNAP-authorized stores, 100% reached or exceeded three varieties, and 96.2% reached or exceeded seven varieties in the fruits and vegetables category, with 80.8% reaching or exceeding 10 varieties, indicating the feasibility of a more stringent requirement pertaining to fruits and vegetables than the current requirement of three varieties. Among SNAP-authorized stores, 84.6% reached or exceeded seven varieties in the meat, poultry, and fish category, suggesting the proposed rule towards meat, poultry, and fish varieties would not be extremely untenable. On average, the dairy category had a significantly lower number of varieties compared to the other three categories; fewer than half of the SNAP-authorized stores carried seven dairy varieties, confirming the concerns that meeting the proposed rule would be challenging for the dairy category. Although the proposed rule expanded the definition of “variety” for dairy to having each main ingredient (e.g., cow, goat, almond, rice, soy) by product type (e.g., milk, yogurt, cheese, butter, infant formula) be considered a discrete variety, there was a lower variety of traditional dairy products to begin with compared to the other three categories. Future policies may retain the current rule of three dairy varieties, but should ensure those are nutrient-dense options. Consistent with the Dietary Guidelines for Americans, cream cheese, cream, and butter should be considered ineligible varieties, as they are not counted toward dairy intake in the Dietary Guidelines for Americans [21]. Additional investigation to inform the definition of eligible varieties within dairy staples and to identify barriers to stocking dairy products is warranted.

### 4.2. Perishability

In this study, 92.3% of all retailers met the perishability requirements under all the rules. Similarly, a study by Powell et al. which used the same audit tool and found that among 113 small retailers in Chicago found that 94.6% were compliant with the original and final rule and 93.8% met the proposed rule [53]. These findings, similar to an analysis from FNS that estimated 98% of small SNAP-authorized stores already stock sufficient perishable foods required in the proposed rule [31], suggest that the concern of insufficient space for perishable foods was not well-founded. This study explored perishability within categories, although it was not proposed in the revisions of the SNAP inventory policy. Limited perishable varieties within categories could be related to several barriers. A report by Frazao et al. suggests that SNAP participants, when provided with additional SNAP dollars, fail to increase spending and consumption on fruits and vegetables [54]. Bodor et al. also pointed out the limited shelf space in small retail stores, which may require a reduction of certain non-perishable items in exchange for perishable foods, potentially leading to profit loss [33].

### 4.3. Depth of Stock

Most retailers in this study carried three or more units of each variety, which is in favor of future expansion of depth-of-stock requirements. Future studies could investigate the practicality of the depth-of-stock requirement of 6 under the first-proposed rules, as it was not assessed in this study. The expansion of depth-of-stock could be an important strategy in improving dietary intake, informed by previous evidence. For example, a one-unit increase in the availability of varieties in fruits and vegetables in corner stores was respectively associated with a 12% and 15% increase in customers’ likelihood to purchase fruits and vegetables [26].

### 4.4. Total Stocking

Most stores (80.0%) met the final rule that required 36 staple food items, particularly SNAP-authorized stores (88.5%). Although only 38.5% of the SNAP-authorized stores met the proposed rule in all four categories (84 items), 73.1% met the proposed rule when the dairy category was excluded (data not shown in Tables). The finding that most retailers were able to meet a stricter rule before the final rule went into effect highlights the potential for small food stores to expand their offerings. This conclusion is consistent with findings by Powell et al. that 81.4% of stores met the variety requirements under the original and final rule, and 22.1% met the variety requirements under the proposed rule, with the least variety in the dairy category (i.e., only 23% of the stores carried at least seven dairy varieties) [53].

Although no disparity was detected between high and low SNAP/high and low racial minority tracts in meeting the different requirements or in perishability within categories, descriptively, in stores with fewer than 10 breads and cereals varieties, 50% of stores in the low SNAP and low racial minority tracts carried >50% perishable varieties in the breads and cereals category, while none of the stores in high SNAP and high racial minority tracts did. It should be noted that non-Hispanic black and Hispanic populations were grouped as high SNAP and high racial minority tracts, potentially masking important differences, as some evidence suggests lower healthful food access and diet quality among non-Hispanic black populations compared to Hispanic populations [4,55,56].

The ability for small retailers to meet the proposed stricter requirement is important in improving food access. Along with various efforts that aim to improve diet quality [57,58], food access serves as a crucial determinant, particularly for residents in geographically isolated locations lacking healthful food outlets [9,59]. Nationally, approximately 80% of SNAP benefits were redeemed in supermarkets and superstores in 2016 [60]. However, evidence from regional studies indicates that SNAP participants with more severe food insecurity shop predominantly at smaller retailers [26], and small retailers have a large share of all SNAP redemptions in low-income neighborhoods, regardless of the presence of supermarkets [23]. Studies of corner stores, gas-marts, dollar stores, and pharmacies have associated increased stocking of healthful foods with increased purchases of healthful foods [25,26,27]. Lower availability of produce was associated with lower purchases of produce and higher purchases of sugar-sweetened beverages in bodegas [28]. Therefore, interventions in small retailers where choices are limited may be a promising solution to help improve the diet quality of low-income populations.

Previous research has identified important challenges that might hinder adequate healthful food provision among small retailers. For example, in a study by Ross et al., store owners identified barriers to expanding food offerings, including low customer demand, high amounts of potential spoilage, and unfair pricing at the wholesaler [34]. Ross et al. suggest that if the proposed rule were to be considered again, the requirements could be revised proportionally to the size of the store and that administrative support and oversight from the USDA should be increased, as they are currently lacking. In particular, stores were largely unaware of the revisions in inventory requirements [34]. Nevertheless, a study by Haynes-Maslow et al. points out that the revision of the inventory rule for the Women, Infants, and Children Program in 2009 raised similar concerns, but the new rule has partly contributed to improved diet quality of its participants, and few stores dropped out due to their inability to meet the inventory requirements [35].

To address challenges in healthful food provision, many innovative strategies have been developed and implemented successfully at the local level, with positive evaluations by stakeholders. These include the Food Trust’s Healthy Corner Store Initiative, ChangeLab Solutions Healthy Small Food Retailer Certification Program, Shop Healthy NYC, Baltimore Healthy Stores, Wholesome Wave, and BrightSide [61,62,63]. Successful interventions have included incentives, improvement of store capacity of stocking and marketing healthful foods, provision of training and technical assistance for store owners, connecting store owners with local partners to encourage healthful food sourcing, and produce delivery to stores [61,62,63]. Additionally, small retailers that struggle with sustaining adequate food varieties may consider cyclical inventory strategies to optimize flexibility of food provisions as seasons shift.

Some limitations of the present study must be noted. First, data collected for this study were cross-sectional and reflected food availability in stores only during the audit timeframe. No qualitative data were collected to reflect perceived food availability and feasibility of expansion from stores’ perspectives, which might require further investigation. Additionally, pricing of food items was not assessed, but is a prioritized factor in food choice among low-income populations [64,65]. Finally, identification of food deserts based on census tract boundaries does not necessarily reflect food shopping behavior. Future studies may consider a multifactorial approach to identifying food desert areas, such as considering store proximity and walkability [66]. Though this study only measured food store environments and did not investigate its association with purchase behavior and the dietary intake of customers, many previous studies have associated low provision of healthful foods with low diet quality, and improved provision with increased purchasing and intake [26,67,68].

Several strengths of this study should be highlighted as well. First, data were collected prior to implementation of the final rule, allowing for assessment of the prior inventory status. Census tracts were carefully selected, aiming for specific characteristics of interest in order to reflect the demographic composition and healthful food access among small retailers in Providence. A broad sample coverage of 73% of the small retailers helped reduce risk for selection bias so that the findings were representative of the actual environment, and increased the generalizability of the findings to other low-income low-access communities as defined in this study, in Providence and cities with a similar demographic makeup and SNAP participation rate. Additional studies in rural areas are needed to demonstrate if similar conclusions can be applied.

Further investigations among SNAP-authorized small retailers are warranted to help inform the efficacy and feasibility of the extension of the perishability requirement within categories, and if existing efforts at the local level could be expanded to a larger scale to support retailers in meeting the extension. Active communication between policy enforcement agencies and store owners are required to ensure sustainable operations, while moving towards the improvement of food accessibility.

## 5. Conclusions

During November and December of 2018, most SNAP-authorized stores (88.5%) sampled from a selected LILA area in Providence, RI were already compliant with the final rule prior to its implementation, and 73.1% already met the proposed minimum of seven varieties in three categories other than dairy. Future expansion of inventory requirements could potentially lead to increased healthful food availability without imposing undue burdens on small retailers in Providence, RI, excluding increased requirements for the dairy variety. If expansions of the requirements were to occur, stakeholders should draw on insights from existing initiatives to assist in a successful implementation. Other initiatives looking to improve healthful food access should borrow current knowledge from SNAP and balance the competing interest among consumers, agricultural sectors, and food distribution sectors [69]. The findings of this study advocate for the future expansion of SNAP inventory requirements, which may be one important strategy in improving healthful food availability and help to facilitate the improvement of diet quality of food purchases from small retailers in food deserts.

## Figures and Tables

**Figure 1 ijerph-16-01052-f001:**
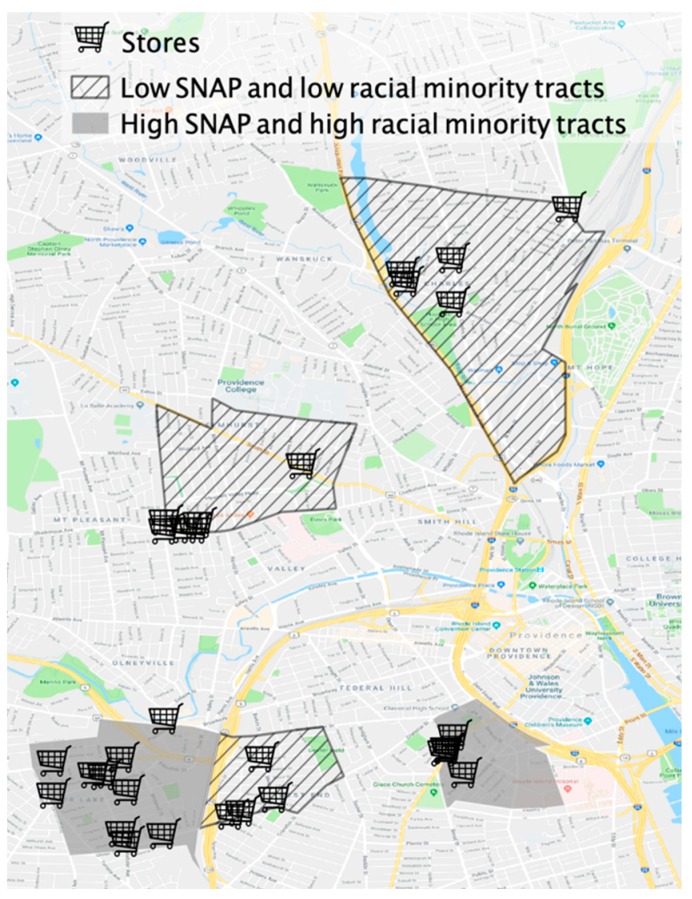
Map of Providence with 15 stores in high SNAP and high racial minority tracts ^1^ and 15 stores in low SNAP and low racial minority tracts ^2^. ^1^ Tracts where the mean of the proportion of black and Hispanic residents and SNAP participating households are 31.5% and 37.6%, respectively. ^2^ Tracts where the mean of the proportion of black and Hispanic residents and SNAP participating households are 2.8% and 3.1%, respectively.

**Table 1 ijerph-16-01052-t001:** Characteristics of five selected tracts in Providence, Rhode Island (RI).

	Tract	Black or African American (%) Beyond 1/2 Mile from Supermarket	Hispanic or Latino (%) Beyond 1/2 Mile from Supermarket	Housing Units Receiving SNAP (%) Beyond 1/2 Mile from Supermarket	Poverty Rate (%)	Housing Units Without Vehicle and Beyond 1/2 Mile from Supermarket (%)
Low SNAP and low racial minority tracts ^1^	44,007,002,300	0.91	1.56	1.61	26.3	1.43
	44,007,002,900	1.43	2.00	2.17	27.7	1.67
	44,007,001,300	1.47	9.24	5.36	29.5	1.67
High SNAP and high racial minority tracts ^2^	44,007,001,600	8.88	50.42	30.42	27.8	6.6
	44,007,000,700	33.86	32.99	44.7	50.2	30.8

^1^ Low proportion of non-Hispanic black and Hispanic (mean = 2.8%), and low proportion of housing units receiving SNAP benefits (mean = 3.1%). ^2^ High proportion of non-Hispanic black and Hispanic (mean = 31.5%), and high proportion of housing units receiving SNAP benefits (mean = 37.6%).

**Table 2 ijerph-16-01052-t002:** Number of stores by store type in high and low Supplemental Nutrition Assistance Program (SNAP)/high and low racial minority tracts (*n* = 30).

	High ^1^	Accept SNAP	Low ^2,^*	Accepts SNAP	Total
Non-chain grocery	9	8	3	3	12
Convenience store	4	3	8	7	12
Small Discount Store	0	0	1	1	1
Drug Store/Pharmacy	2	2	2	2	4
Liquor Store	0	0	1	0	1

* Includes three stores (two non-chain grocery and one small discount store) that do not fall within the low SNAP and low racial minority tracts but are located at the boundary line of tract 44,007,002,300. The stores were added to achieve more comprehensive sampling in low SNAP and low racial minority tracts. ^1.^ Tracts where the mean of the proportion of black and Hispanic residents and SNAP participating households are 31.5% and 37.6%, respectively. ^2.^ Tracts where the mean of the proportion of black and Hispanic residents and SNAP participating households are 2.8% and 3.1%, respectively.

**Table 3 ijerph-16-01052-t003:** Proportion of stores meeting the different versions of variety, perishable categories, depth of stock, and total stocking requirements (*n* = 30).

	Original Rule (%)				Proposed Rule (%)				Final Rule (%)			
	All Stores	High ^1^	Low ^2^	*p* Value	All Stores	High ^1^	Low ^2^	*p* Value	All Stores	High ^1^	Low ^2^	*p* Value
Variety ^3^	83.3	86.7	80	1	33.3	26.7	40	0.7	83.3	86.7	80	1
Perishable categories ^4^	83.3	80	86.7	1	83.3	80	86.7	1	83.3	80	86.7	1
Depth of stock ^5^	100	100	100		96	93.3	100	1	96	93.3	100	1
Total stocking ^6^	80	80	80		33.3	26.7	40	0.7	80	80	80	
	SNAP Authorized stores (*n* = 26)	High	Low	*p* value	SNAP Authorized stores (*n* = 26)	High	Low	*p* value	SNAP Authorized stores (*n* = 26)	High	Low	*p* value
Variety ^3^	92.3	92.3	92.3		38.5	30.8	46.2	0.688	92.3	92.3	92.3	
Perishable categories ^4^	92.3	84.6	100	0.48	92.3	84.6	100	0.48	92.3	84.6	100	0.48
Depth of stock ^5^	100	100			96.1	92.3	100	1	96.1	92.3	100	1
Total stocking ^6^	88.5	84.6	92.3	1	38.5	30.8	46.2	0.688	88.5	84.6	92.3	1

^1^ Tracts where the mean of the proportion of black and Hispanic residents and SNAP participating households are 31.5% and 37.6%, respectively. ^2^ Tracts where the mean of the proportion of black and Hispanic residents and SNAP-participating households are 2.8% and 3.1%, respectively. ^3^ “Variety” refers to different types of foods. For example, apples, cabbage, and squash in the fruits and vegetables category would be variety of 3. The following does not meet the variety requirement: having different brands and sizes; having the same item but with varying ingredients (e.g., plain sausage and spicy sausage); or having the same item but offering different types of that item (e.g., Granny Smith and Red Delicious apples). Minimum requirements under the original rule, proposed rule, and final rule are 3, 7, and 3, respectively. ^4^ “Perishable categories” refers to categories with perishable items- that is, items that are either frozen staple food items, or fresh, un-refrigerated, or refrigerated staple food items that will spoil or suffer significant deterioration in quality within 2 to 3 weeks. Minimum requirements under the original rule, proposed rule, and final rule are 2, 3, and 2, respectively. ^5^ “Depth of Stock” refers to the minimum number of units of every variety; that is, if the depth of stock is 3, stores must have at least three units per variety. For example, if apples are going to be counted in the fruits and vegetables category, at least three apples and/or products with the apple as the first ingredient, such as applesauce or apple juice, need to be stocked in the store. Food items that are in storage and not available on the shelves at the time of auditing were not counted. Minimum requirements under original rule, proposed rule, and final rule are 1, 3, and 3, respectively. ^6^ “Total stocking” considers the category, variety, and depth of stock. For example, seven varieties in each category with a depth of stock of 3 results in a total stocking of 84 (i.e., 4 categories × 7 varieties × 3 depth of stock). Minimum requirements under the original rule, proposed rule, and final rule are 12, 84, and 36, respectively.

**Table 4 ijerph-16-01052-t004:** Mean of variety, variety of perishable items, depth of stock, and total stocking in each category (*n* = 30).

	Variety (mean ± SD)				Variety of Perishable Items (mean ± SD)				Depth of Stock				Total Stocking			
	All stores	High ^1^	Low ^2^	*p*-value	All stores	High ^1^	Low ^2^	*p*-value	All stores	High ^1^	Low ^2^	*p*-value	All stores	High ^1^	Low ^2^	*p*-value
Fruits and Vegetables	8.53 ± 3.16 ^a^	8.60 ± 3.18	8.46 ± 3.25	0.910	6.20 ± 4.09 ^c^	6.20 ± 4.14	6.20 ± 4.18	/	3.94 ± 0.31 ^e^	3.88 ± 0.44 ^e^	4.00 ± 0.00 ^e^	0.326	33.62 ± 12.63 ^f^	33.9 ± 13.08 ^f^	33.33 ± 12.62 ^f^	0.904
Meat Poultry and Fish	7.83 ± 3.06 ^a^	7.80 ± 2.78	7.87 ± 3.42	0.954	4.67 ± 2.58 ^c^	4.53 ± 2.72	4.80 ± 2.51	0.782	4.00 ± 0.00 ^e^	4.00 ± 0.00 ^e^	4.00 ± 0.00 ^e^		31.20 ± 12.13 ^f^	31.20 ± 11.13 ^f^	31.20 ± 13.45 ^f^	1.000
Breads and Cereals	7.80 ± 3.32 ^a^	7.87 ± 3.09	7.73 ± 3.63	0.915	2.07 ± 1.70 ^d^	1.60 ± 1.45	2.53 ± 1.85	0.136	3.98 ± 0.14 ^e^	3.95 ± 0.19 ^e^	4.00 ± 0.00 ^e^	0.326	30.84 ± 13.48 ^f^	30.74 ± 12.84 ^f^	30.93 ± 14.54 ^f^	0.970
Dairy	5.40 ± 3.04 ^b^	5.27 ± 3.24	5.53 ± 2.92	0.815	4.80 ± 2.54 ^c^	4.60 ± 2.61	5.00 ± 2.54	0.674	4.00 ± 0.00 ^e^	4.00 ± 0.00 ^e^	4.00 ± 0.00 ^e^		21.47 ± 11.91 ^f^	20.80 ± 12.49 ^f^	22.13 ± 11.70 ^f^	0.765
All categories	29.57 ± 11.39	29.53 ± 10.82	29.60 ± 12.32	0.988	17.73 ± 8.49	16.93 ± 8.89	18.53 ± 8.29	0.614	3.96 ± 0.18 ^e^	3.93 ± 0.26 ^e^	4.00 ± 0.00 ^e^	0.326	117.12 ± 45.66 ^f^	116.65 ± 44.06^f^	117.60 ± 48.75 ^f^	0.956
	Variety (mean ± SD)			*p*-value	Variety of perishable items (mean ± SD)			*p*-value	Depth of stock			*p*-value	Total stocking			*p*-value
SNAP-authorized stores (*n* = 26)	33.15 ± 6.21			0.010 *	20.04 ± 6.31			0.001 **	3.96 ± 0.20 ^e^			0.702	131.30 ± 25.80 ^f^			0.000 ***
Non-SNAP-authorized stores (*n* = 4)	6.25 ± 9.84				2.75 ± 4.27				4.00 ± 0.00 ^e^				25.00 ± 39.38 ^f^			

^a–b^ Means within a column with different superscripts differ (*p* < 0.05). Homogeneity of variances was not violated. ^c–d^ Means within a column with different superscripts differ. Homogeneity of variances was violated. Welch F (3, 62.079) = 15.772 (*p* = 0.000). ^e^
*n* = 4 was assigned to a variety of 3 or more. ^f^ Calculated based on estimated and assigned variety and depth of stock for the purpose of comparison between tracts, not to be taken literally. ^1^ Tracts where the mean of the proportion of black and Hispanic residents and SNAP-participating households are 31.5% and 37.6%, respectively.^2^ Tracts where the mean of the proportion of black and Hispanic residents and SNAP-participating households are 2.8% and 3.1%, respectively. * significant at *p* < 0.05; ** significant at *p* < 0.005; *** significant at *p* < 0.001.

**Table 5 ijerph-16-01052-t005:** Estimate of the proportion of perishable items out of the total items recorded in each category.

	≥1 Perishable Item (in %, *n* = 30)				>50% Perishable Items							
					All Stores (in %, *n* = 30)				Stores with < 10 Items in Each Category (in %)			
	All stores	High ^1^	Low ^2^	*p* value	All stores	High ^1^	Low ^2^	*p* value		High ^1^	Low ^2^	*p* value
Fruits and Vegetables	90	86.7	93.3	1	73.3 *	73.3 *	73.3 *		55.6 (*n* = 5/9 stores)	75	40	0.524
Meat, Poultry, and Fish	90	93.3	86.7	1	76.7 *	66.7 *	86.7 *	0.39	70.6 (*n* = 12/17 stores)	70	71.4	1
Breads and Cereals	80	73.3	86.7	0.651	16.7 *	6.7 *	26.7 *	0.33	21.4 (*n* = 3/14 stores)	0	50	0.055
Dairy	86.7	86.7	86.7		86.7 *	86.7 *	86.7 *	1	86.2 (*n* = 25/29 stores)	85.7	86.7	1
All categories	76.7	66.7	86.7	0.39	10 *	0	20 *	0.224				

* As per the instructions, a maximum of 10 varieties were recorded for each category during store auditing. Stores that were recorded to have met 10 varieties were likely to have carried more than 10 varieties, so the proportions of perishable items in those stores were estimates. ^1^ Tracts where the mean of the proportion of black and Hispanic residents and SNAP participating households are 31.5% and 37.6%, respectively. ^2^ Tracts where the mean of the proportion of black and Hispanic residents and SNAP participating households are 2.8% and 3.1%, respectively.

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
