# Peer review of "Staple Food Item Availability among Small Retailers in Providence, RI"

_ijerph, 2019, doi:10.3390/ijerph16061052_

Round 1
Reviewer 1 Report
This research addresses a topic that is timely and likely of interest to readers of IJERPH. The authors investigate the feasibility of current and proposed inventory requirements for authorized SNAP retailers. The research topic has important policy implications, and I appreciate the opportunity to review it.
Although the topic is of interest, the manuscript has weaknesses that should be addressed prior to publication. Mainly, there are some missing details and other clarity issues that need attention, particularly in the discussion.
Specific suggestions are listed below, separated by section:
Introduction:
Line 45: The term racially-diverse is unclear. Do you mean there is a non-white majority or that here are a variety of races and/or ethnicities in these neighborhoods?
2. Line 66 (and table footnotes): Define stocking depth and avoid using the term depth of stock in the definition.
3. The inventory requirements for SNAP retailers are not mentioned until line 61. You might consider introducing the topic earlier. Ideally the reader wouldn't have to read so much background before being introduced to the paper topic.
Methods:
4. Lines 128-132 and Table 1: The Low SNAP and racial minority tracts and high SNAP and racial minority tracts categories are confusing. It reads as if the adjective, low or high, is applied to both SNAP and racial minority tracts. Consider categorizing them as racial minority tracks with high snap and racial minority tracks with low SNAP.
5. Table 2: High and Low in the table header is insufficient. More detail needed.
6. Figure 1: The figure does not stand alone. More detail is needed, including definitions of the two tract categories.
7. Line 163: Were auditors instructed to do the variety assessment in a systematic order? If not, how likely is that certain varieties were systematically assessed first, due to ease, leaving other varieties less likely to be captured in the audit?
8. Lines 171- 172: Was this audit tool limitation addressed in the discussion section? How may it have influenced your results?
9. Line 172: Was this webinar live or taped? If taped, how were protocol questions handled?
10. Were the audits unannounced? Please specify.
11. Lines 201-201: Depth of stock concept is confusing. I assume that it means what is available on the shelf during the audit. Consider specifying whether items in the back of the store (i.e. stock) are included.
12. Lines 204-205: The total stocking criteria is also confusing. Consider providing an example of the math required to generate the 12, 36, or 84 total items.
Results:
13. Be consistent in the way results are presented. For example, in lines 213-222, no n value is shared, but then it is in line 222 and elsewhere. I'm not sure what the n=25 value refers to in line 222.
Discussion:
14. The first part of the discussion is very choppy. I suggest adding more meat to the first few lines (296-298) so that the section begins with a summative declaration of the top results.
15. Lines 312-315: This discussion on cream cheese, etc. seems to come out of nowhere. Consider providing more context here or in the introduction.
16. Lines 345-347: These seem to be key results. Why are they buried so deep into the discussion?
17. Line 351: Is there are a word missing after Hispanic?
18. Lines 380-382: Do you have any references for these programs?
19. Line 384: Do the partners both encourage healthful food sourcing and deliver produce to the stores? Perhaps the Oxford comma is missing here, but it is present elsewhere. So, I am unsure of the sentence meaning.
20. Line 390: Consider providing more detail about the desired multifactorial appraoch.
21. Lines 400-401: Discuss the generalizability of your findings to retailers outside of Providence.
Conclusion:
22. This paragraph was hard to follow. In particular, lines 414-418 are unclear. Consider re-writing these to be more direct/active and succinct sentences.
23. Line 420: There is no evidence that the provision of healthy food alone will improve diet quality. Perhaps it may facilitate diet quality improvements. This language should be softened accordingly.
Author Response
Response to Reviewer 1 Comments
Introduction:
1: Line 45: The term racially-diverse is unclear. Do you mean there is a non-white majority or that here are a variety of races and/or ethnicities in these neighborhoods?
Response 1: We have clarified the term “racially-diverse” to reflect neighborhoods with predominantly Black and Hispanic populations (line 45-46).
2. Line 66 (and table footnotes): Define stocking depth and avoid using the term depth of stock in the definition.
Response 2: We have changed “stocking depth” to “depth of stock” (line 74 and 87), added definition of “depth of stock” as “the minimum number of units of every variety” in line 75, and revised this definition in the footnote of Table 3 (line 306) for clarity.
3. The inventory requirements for SNAP retailers are not mentioned until line 61. You might consider introducing the topic earlier. Ideally the reader wouldn't have to read so much background before being introduced to the paper topic.
Response 3: We agree with the reviewer that SNAP inventory requirements can be introduced earlier and we now make mention of it when SNAP was first introduced (line 52-53): “Recently, SNAP proposed expanding stores’ inventory requirements in an effort to improve healthful food access for low-income populations, among whom 42 million participate in SNAP.”
Methods:
4. Lines 128-132 and Table 1: The Low SNAP and racial minority tracts and high SNAP and racial minority tracts categories are confusing. It reads as if the adjective, low or high, is applied to both SNAP and racial minority tracts. Consider categorizing them as racial minority tracks with high snap and racial minority tracks with low SNAP.
Response 4: We agree with the reviewer that this reads unclearly. We have changed “high SNAP and racial minority tracts” to “high SNAP and high racial minority tracts”; “low SNAP and racial minority tracts” to “low SNAP and low racial minority tracts”; and “high and low SNAP and racial minority tracts” to “high and low SNAP /high and low racial minority tracts” throughout the manuscript to avoid confusions from wording.
5. Table 2: High and Low in the table header is insufficient. More detail needed.
Response 5: We have clarified the table header (Line 157) as “Number of stores by store type in high and low SNAP /high and low racial minority tracts”.
We have also added a footnote of “1. Tracts where the mean of the proportion of Black and Hispanic residents and SNAP participating households are 31.5% and 37.6% respectively.” and
“2. Tracts where the mean of the proportion of Black and Hispanic residents and SNAP participating households are 2.8% and 3.1% respectively.” to clarify what “High” and “Low” refers to under Table 2,3,4,5 to allow these tables to stand alone.
6. Figure 1: The figure does not stand alone. More detail is needed, including definitions of the two tract categories.
Response 6: We have added the footnote of “1. Tracts where the mean of the proportion of Black and Hispanic residents and SNAP participating households are 31.5% and 37.6% respectively.” And “2. Tracts where the mean of the proportion of Black and Hispanic residents and SNAP participating households are 2.8% and 3.1% respectively.” to clarify what “High” and “Low” refers to under Figure 1 (line 169-175). We have also clarified this information in the Method section in line 136-141.
7. Line 163: Were auditors instructed to do the variety assessment in a systematic order? If not, how likely is that certain varieties were systematically assessed first, due to ease, leaving other varieties less likely to be captured in the audit?
Response 7: We thank the reviewer for pointing out this missing information in the methodology. It is now included in line 182-185 as “Auditors recorded information on up to 10 foods and/or beverage varieties under each of the four staple food categories, in the order from Fruits and Vegetables category to Meat, Poultry, Fish category to Breads and Cereals category and lastly to Dairy category. Under each category, food items were searched for from left to right in the sequence shown on the instrument page, giving priority to perishable items.”
8. Lines 171- 172: Was this audit tool limitation addressed in the discussion section? How may it have influenced your results?
Response 8: We thank the reviewer for this addition and we have included it into the discussion in line 377-378 as “Future studies could investigate the practicality of the depth of stock requirement of 6 under the first proposed rules, as it was not assessed in this study.”
9. Line 172: Was this webinar live or taped? If taped, how were protocol questions handled?
Response 9: The webinar was live and taped. We have added this description in line 198-200 as “The webinar was taped for reference and any questions were answered either during the webinar or afterwards via email communication.”
10. Were the audits unannounced? Please specify.
Response 10: Yes. It is now specified in line 203 as “Data collection was unannounced, but an explanatory letter written in English and Spanish was provided to store owners and staff when concerns were raised.”
11. Lines 201-201: Depth of stock concept is confusing. I assume that it means what is available on the shelf during the audit. Consider specifying whether items in the back of the store (i.e. stock) are included.
Response 11: We agree that this is important information to disclose. We have made this addition into the Method section (line 188-189) and in footnote of Table 3 (line 314-315).
12. Lines 204-205: The total stocking criteria is also confusing. Consider providing an example of the math required to generate the 12, 36, or 84 total items.
Response 12: The example is now provided as “7 varieties in each category with depth of stock of 3 results in total stocking of 84 (i.e. 4 categories x7 varieties x3 depth of stock).” This was edited in the footnote (line 316-317), and was added to the introduction as well (line 76).
Results:
13. Be consistent in the way results are presented. For example, in lines 213-222, no n value is shared, but then it is in line 222 and elsewhere. I'm not sure what the n=25 value refers to in line 222.
Response 13:
We have deleted the numbers, leaving just the percentages for consistency and clarity.
Discussion:
14. The first part of the discussion is very choppy. I suggest adding more meat to the first few lines (296-298) so that the section begins with a summative declaration of the top results.
Response 14: We have included a brief summary in line 337-343 according to the reviewer’s suggestion: “Overall, most stores met all requirements including variety, perishability, and depth of stock in the original and final rules, and a higher number of stores would have also met all requirements in the proposed rule if the dairy variety requirement was excluded. No disparity was observed between high and low SNAP /high and low racial minority tracts in meeting the different requirements in all categories in the original, final and proposed rule or in perishability within categories. Detailed findings are discussed below.”
15. Lines 312-315: This discussion on cream cheese, etc. seems to come out of nowhere. Consider providing more context here or in the introduction.
Response 15: We agree that more context was needed and have included a brief discussion in the introduction (line 66-70): “According to USDA Food and Nutrition Service (FNS), staple foods are defined as basic food items that make up a significant portion of an individual’s diet and are usually prepared at home and consumed as a major component of a meal. This broad definition allowed for certain food items that could be regarded as accessory foods to be counted towards staple foods, for example, cream cheese, cream and butter.”
16. Lines 345-347: These seem to be key results. Why are they buried so deep into the discussion?
Response 16: We have included one sentence in the beginning paragraph of the discussion (line 340-343) to highlight this finding.
17. Line 351: Is there are a word missing after Hispanic?
Response 17: Yes, we have added the word “populations” after “Hispanic,” which had been missing previously (line 401).
18. Lines 380-382: Do you have any references for these programs?
Response 18: We have added references for the programs described in line 433.
19. Line 384: Do the partners both encourage healthful food sourcing and deliver produce to the stores? Perhaps the Oxford comma is missing here, but it is present elsewhere. So, I am unsure of the sentence meaning.
Response 19: We thank the reviewer for pointing out the grammatic error and made changes in line 436.
20. Line 390: Consider providing more detail about the desired multifactorial appraoch.
Response 20: To provide more details, this sentence has been modified to “Future studies may consider a multifactorial approach to identify food desert areas, for example, considering store proximity and walkability.”
21. Lines 400-401: Discuss the generalizability of your findings to retailers outside of Providence.
Response 21: The generalizability of our findings was described as “to other low-income low-access communities as defined in this study, in Providence and cities with similar demographic makeup and SNAP participation rate” (line 456-457). “Additional studies in rural areas are needed to demonstrate if similar conclusions can be applied.” was added to address generalizability in rural areas in line 458.
Conclusion:
22. This paragraph was hard to follow. In particular, lines 414-418 are unclear. Consider re-writing these to be more direct/active and succinct sentences.
Response 22: We have re-written this sentence according to the reviewer’s suggestion (line 471-475): “If expansions of the requirements were to occur, stakeholders should draw on insights from existing initiatives to assist in successful implementation. Other initiatives looking to improve healthful food access should borrow current knowledge from SNAP and balance the competing interest among consumers, agricultural sectors and food distribution sectors.”
23. Line 420: There is no evidence that the provision of healthy food alone will improve diet quality. Perhaps it may facilitate diet quality improvements. This language should be softened accordingly.
Response 23: We have softened this conclusion in line 479-482 to “Findings of this study advocate for future expansion of SNAP inventory requirements, which may be one important strategy in improving healthful food availability and help to facilitate improvement of diet quality of food purchases in small retailers in food deserts.”

Reviewer 2 Report
Thank you for the opportunity to review. I regard this paper to be of significant merit, presenting - as it does - a good news story about the availability of food products, irrespective of location in terms of low income, low access and/or minority populations. It successfully dispels important existing myths to this effect. I consider this paper to be worthy in terms of its account of fruit and vegetables' availability in light of the final, original and proposed rules re: perishability and stocking densities. I consider that this paper contributes to the relevant knowledge base and merits publication.
I believe that in the Strengths and Limitations section it would be prudent to itemize how no qualitative exploration was conducted and that the authors could signpost usefully the need for further research investigating the barriers to stocking dairy products, and additional qualitative research with retailers in respect of the pragmatism of more stringent stocking inventory requirements. I also believe that the discussion could include some articulation of recommendations around cyclical variety of food products to promote further dietary and nutritional quality through varied stocking patterns, and avoid potential for reduced variety of food product categories on offer.
Author Response
Response to Reviewer 2 Comments
1. I believe that in the Strengths and Limitations section it would be prudent to itemize how no qualitative exploration was conducted and that the authors could signpost usefully the need for further research investigating the barriers to stocking dairy products, and additional qualitative research with retailers in respect of the pragmatism of more stringent stocking inventory requirements.
Response 1:
We thank the reviewer for the addition of these important discussions. We have included the sentence “Additional investigation to inform the definition of eligible varieties within dairy staples and to identify barriers to stocking dairy products is warranted.” in line 360-361 and the sentence “No qualitative data were collected to reflect perceived food availability and feasibility of expansion from stores’ perspectives, which might require further investigations.” in line 441-442.
2. I also believe that the discussion could include some articulation of recommendations around cyclical variety of food products to promote further dietary and nutritional quality through varied stocking patterns, and avoid potential for reduced variety of food product categories on offer.
Response 2:
We agree with the reviewer’s suggestion to acknowledge cyclical inventory as a pragmatic strategy, and have included this recommendation in line 436-438: “Additionally, small retailers that struggle with sustaining adequate food varieties may consider cyclical inventory strategies to optimize flexibility of food provisions as seasons shift.”
